# Dualistic Effects of PRKAR1A as a Potential Anticancer Target in Cancer Cells and Cancer-Derived Stem Cells

**DOI:** 10.3390/ijms25052876

**Published:** 2024-03-01

**Authors:** Joong-Won Baek, A-Reum Nam, Kyunggon Kim, Pyung-Hwan Kim

**Affiliations:** 1Department of Biomedical Laboratory Science, Konyang University, Daejeon 35365, Republic of Korea; wkdqordnjs55@gmail.com; 2Department of Veterinary Medicine, Seoul National University, Seoul 08826, Republic of Korea; arbjlvz@snu.ac.kr; 3Department of Convergence Medicine, Asan Medical Center, Seoul 05505, Republic of Korea

**Keywords:** cancer, cancer stem cells, proteomic analysis, new biomarkers, dualistic effect

## Abstract

The integration of innovative medical technologies and interdisciplinary collaboration could improve the treatment of cancer, a globally prevalent and often deadly disease. Despite recent advancements, current cancer therapies fail to specifically address recurrence and target cancer stem cells (CSCs), which contribute to relapse. In this study, we utilized three types of cancer cells, from which three types of CSCs were further derived, to conduct a proteomic analysis. Additionally, shared cell surface biomarkers were identified as potential targets for a comprehensive treatment strategy. The selected biomarkers were evaluated through short hairpin RNA treatment, which revealed contrasting functions in cancer cells and CSCs. Knockdown of the identified proteins revealed that they regulate the epithelial–mesenchymal transition (EMT) and stemness via the ERK signaling pathway. Resistance to anticancer agents was consequently reduced, ultimately enhancing the overall anticancer effects of the treatment. Additionally, the significance of these biomarkers in clinical patient outcomes was confirmed using bioinformatics. Our study suggests a novel cancer treatment strategy that addresses the limitations of current anticancer therapies.

## 1. Introduction

Cancer is a globally prevalent disease that accounts for approximately 25% of all deaths in most developed countries, making it the leading cause of death worldwide [1,2]. It can metastasize to other tissues and organs, which makes it difficult to cure [3]. Cancer-derived stem cells (CSCs) constitute a small population of cancer cells and are important in cancer treatment. This cell population accounts for 1–2% of cancer tissue and plays a pivotal role in cancer recurrence post treatment.

However, unlike normal stem cells, they can initiate tumor formation (tumorigenicity), propagate tumors throughout the body, and undergo unlimited self-renewal [4,5]. Typically existing in a dormant state, CSCs demonstrate resistance to anticancer drugs. This poses a formidable obstacle to their eradication during cancer therapy and considerably contributes to cancer relapse and metastasis [6]. Consequently, comprehensive cancer treatment requires the targeted removal of both cancer cells and CSCs.

In recent years, research into targeted cancer therapies for CSCs has become increasingly active, complementing conventional surgical procedures, radiation therapy, and chemotherapy [7]. Nevertheless, challenges persist because of the involvement of CSCs in diverse signaling pathways, limited knowledge regarding substances targeting this small CSC population, and the high resistance of these cells to anticancer agents [8].

A new approach to cancer treatment is required to overcome these limitations. Early cancer detection and appropriate treatment can substantially enhance the 5-year survival rate by >80–90%. Moreover, early diagnosis facilitates the development of feasible treatment options and interventions [9,10]. Therefore, technological advancements have enabled the discovery of biomarkers using various omics-based analyses for early cancer detection [11,12].

New biomarkers, indicative of physiological changes based on macromolecules such as DNA, RNA, and proteins within the human body, have been utilized for prognostic confirmation after disease treatment or for the diagnosis of early-stage cancer [13]. However, the discovery of new markers is challenging, owing to the heterogeneous nature of cancer. Therefore, the variability in treatment effectiveness due to genetic differences and the simultaneous treatment of various types of cancer and CSCs with a single or combination therapeutic approach necessitates the development of innovative cancer therapeutics.

To address this, we utilized proteomic analyses to identify new common biomarkers on the cell surfaces of lung, liver, and breast cancer and in each cancer cell-derived CSC. The incidence of these cancer types is expected to increase due to several factors. For example, the recent pandemic continues to increase the prevalence of lung cancer, whereas the escalating worldwide prevalence of obesity, attributed to the abundance of food, is associated with an increasing incidence of liver cancer. Furthermore, the consistently high incidence and mortality rate of breast cancer in women remains a major challenge [14,15,16]. Therefore, the aim of this study was to utilize proteomic analyses to identify novel biomarkers in both cancer cells and CSCs. This study further aimed to evaluate the biological functions of these biomarkers within each cell type and assess their potential as novel biomarkers.

## 2. Results

### 2.1. PRKAR1A Expression at the Transcriptomic Level of Various Cell Lines

To discover novel cell surface biomarkers, a proteomic analysis was conducted on six different cell types to identify common biomarkers expressed in the three types of cancer cells and CSCs examined. A total of 7243 proteins were examined, with a focus on 4812 cell surface proteins, considering their potential as target substances and diagnostic biomarkers.

Of the selected 4812 proteins, 427 (8.9%) were consistently highly expressed in lung, liver, and breast cancer cells (Figure 1A). Of these, protein kinase cAMP-dependent type Ⅰ regulatory subunit alpha (PRKAR1A) emerged as a top candidate biomarker. Despite its potential applications as a diagnostic and therapeutic agent for cancer, it remains largely underexplored. This gene is associated with the induction of the Carney complex when mutated and has unclear functions in cancer [17]. To validate the expression levels of PRKAR1A in cancer cells and CSCs, RNA was extracted from each cell type, the cDNA was synthesized, and conventional PCR was performed to confirm its expression at the transcriptomic level (Figure 1B). The cell lines used included human lung cancer A549 cells, A549-derived lung cancer stem cells (LCSC), human liver cancer Huh7 cells, Huh7-derived liver cancer stem cells (LiCSC), human breast cancer MCF7 cells, and MCF7-derived breast cancer stem cells (BCSC). Additionally, umbilical cord blood-derived mesenchymal stem cells (UCB-MSC) were used as the control cells of the cell lines, used to compare PRKAR1A expression.

As shown in Figure 1B, PRKAR1A was not expressed in the control UCB-MSCs. However, consistent with the results of proteomic analysis, PRKAR1A was commonly expressed in cancer cells and CSCs. The quantitative data for the detected PRKAR1A band showed a similar tendency, as shown in Figure 1B (Figure 1C). These results indicate that PRKAR1A is expressed in cancerous cell lines whereas it is not expressed in non-cancer cell lines.

### 2.2. Efficient shRNA Selection for Biological Functional Assessment of PRKAR1A

To assess the biological functions of PRKAR1A in cancer cells and CSCs, we generated a series of short hairpin RNAs (shRNA) to downregulate PRKAR1A expression in each cell line. The cells were initially transfected with a PRKAR1A-targeting shRNA series co-expressing GFP in the vector to select the most effective shRNA. The consistent expression of GFP across all the transfected cells verified the successful transfection of shRNA (Appendix A). RNA was then extracted from the cells transfected with shRNAs, the cDNA was synthesized, and the shRNA that most effectively knocked down PRKAR1A expression was selected. In all six cell lines, shRNA-PRKAR1A C (referred to as shC) consistently reduced the expression of PRKAR1A (Figure 2A,B). Similar results were observed from the quantitative data on the detected PCR bands (Figure 2C,D). Based on this result, the data in Figure 2 were obtained.

### 2.3. Biological Functional Assessment through PRKAR1A Knockdown Using shRNA

We assessed the effects of PRKAR1A knockdown on the biological functions of cancer cells and CSCs using shRNA. Although this protein regulates the cAMP signaling pathway, its function in cancer cells remains unclear [17,18,19,20,21]. Therefore, we evaluated the effect of the shRNA-mediated knockdown of PRKAR1A on cell proliferation and migration.

In the cell proliferation assay [22], the downregulation of PRKAR1A expression increased the proliferation of all the cancer cells tested (Figure 3A) but decreased proliferation in CSCs (Figure 3B) in a time-dependent manner. The long-term cell growth, division, and replication capabilities were examined using a colony formation assay. In the cancer cells, the colonies were more abundant in the shC-treated group (Figure 4A). In contrast, fewer colonies were observed in the shC-treated group in CSCs (Figure 4B). The quantitative analysis of crystal violet staining at an absorbance of 570 nm supported these results (Figure 4C,D).

The altered cell proliferation rate in each cell type is thought to be influenced by changes in proteins associated with the cell cycle [23]. Therefore, we observed changes in the protein cyclin D1 associated with cell cycle progression [23,24]. To this end, the expression of cyclin D1 at the transcriptomic level was evaluated, and a flow cytometry-based cell cycle analysis was conducted. As shown in Figure 5A,B, cyclin D1 expression at the transcriptomic level increased in the cancer cells and decreased in the CSCs following shC treatment. The quantitative data of the detected bands showed a similar tendency (Figure 5C,D). This revealed that PRKAR1A regulates cyclin D1 expression during the G1 phase of the cell cycle. A subsequent cell cycle analysis demonstrated changes in the S phase. The length of the S phase increased in cancer cells, whereas that of the G1 and G2 phases decreased (Appendix A). In the CSCs, a decrease in cyclin D1 expression reduced the length of the S phase and increased that of the G1 and G2 phases (Appendix A). Cyclin D1 expression was quantified relative to the GAPDH values (Figure 5C,D), and the cell cycle data are graphically represented for each phase in Figure 5E,F. These results indicate that PRKAR1A regulates entry into the S phase of the cell cycle and inversely affects the proliferation of cancer cells and CSCs. Moreover, this clearly indicates that PRKAR1A is likely associated with signaling pathways related to the cell cycle.

Given the differential effect of PRKAR1A on the migration of various cell types [21], we examined its role in cancer cells and CSCs using a scratch assay. Cell migration was significantly increased in the shC-treated cancer cells at the 12 h time point compared to that in the control group at 0 h (Figure 6A). However, the opposite trend was observed in the CSCs (Figure 6B). A quantitative analysis of the scratch width using Image J version 1.52a confirmed these observations (Figure 6C,D).

In summary, our findings demonstrate that the downregulation of PRKAR1A expression increases cell proliferation in cancer cells but decreases it in CSCs. Furthermore, the downregulation of PRKAR1A expression enhances the migration of cancer cells while suppressing that of CSCs.

### 2.4. Regulation of Functionality via the ERK Signaling Pathway and EMT

Based on our previous data confirming that PRKAR1A regulates cell proliferation and migration, we sought to determine whether these effects were associated with ERK phosphorylation in the extracellular signal-regulated kinase (ERK) signaling pathway, well-known for the representative intracellular signaling pathway [25]. To this end, we evaluated the phosphorylation level of ERK (p-ERK) in the cells treated with an ERK inhibitor (PD98059). PD98059 is an inhibitor of the mitogen-activated protein kinase (MAPK) pathway, specifically recognized for its impact on ERK inhibition, affecting cellular processes such as survival, growth, differentiation, and metabolism [26].

Forty-eight hours after shC transfection to downregulate PRKAR1A expression (Figure 7), a change in p-ERK expression was observed in the shC-treated cells compared to that in the control groups. In the cancer cells, p-ERK expression increased following shC treatment, whereas it decreased in the CSCs (Figure 7A,B). In contrast, no changes in the p-ERK levels were observed in any group when the ERK inhibitor PD98059 was used. Quantification of p-ERK expression using β-actin was similar to that observed by Western blotting (Figure 7C,D). These results indicate that PRKAR1A regulates cellular functions via the ERK signaling pathway.

We examined the ERK phosphorylation-mediated regulatory role of PRKAR1A shRNA in the EMT of cancer cells. EMT is a reversible cellular process that transitions cells from an epithelial (E) to a more mesenchymal (M) state, with the potential for cells in the M state to revert to the E state through mesenchymal–epithelial transition (MET) [27,28]. EMT plays a crucial role in embryonic development, wound healing, and cancer progression and influences the activity of the ERK signaling pathway [29]. Therefore, it is imperative to investigate whether PRKAR1A affects EMT and, ultimately, stemness by regulating ERK phosphorylation. Moreover, the expression of specific genes associated with stemness, such as the sex-determining region Y-box 2 (*sox2*), the octamer-binding transcription factor 4 (*oct4*), and the Nanog homeobox (*Nanog*), can be regulated during EMT [30], potentially enhancing stem cell characteristics, cellular motility, and invasiveness. EMT is involved in mechanisms related to cancer invasion and metastasis, contributing to the formation of CSCs with stem cell properties in the early stages of tumorigenesis [30,31,32].

The expression of the epithelial marker E-cadherin decreased in the cancer cells treated with shC, whereas that of the mesenchymal markers N-cadherin, Snail, Slug, and vimentin increased (Figure 8A) at the transcriptomic level. However, contrasting results were observed in the CSCs (Figure 8B). Expression of these markers was quantified relative to that of GAPDH (Figure 8C,D). This trend was also observed at the protein level when PRKAR1A was knocked down (Figure 8E,F).

EMT changes were observed in the biomarkers related to stemness in the cancer cells and CSCs, such as *sox2*, *oct4*, and *Nanog*, which are associated with stem cell characteristics [29,30,31,32]. The expression of stemness biomarkers increased in cancer cells (Figure 9A), whereas it decreased in CSCs (Figure 9B). The quantification of the stemness biomarker’s expression relative to that of GAPDH supported these findings (Figure 9C,D).

These results demonstrate that PRKAR1A can regulate both EMT and stemness through the ERK signaling pathway, thereby orchestrating distinct regulatory patterns in the biological functions of cancer cells and CSCs, with opposing effects.

### 2.5. Downregulation of PRKAR1A Expression Enhanced Sensitivity to Anticancer Agents

CSCs are inherently resistant to anticancer agents [6], owing to their acquisition of EMT and stemness. Therefore, we evaluated how the downregulation of PRKAR1A expression affects CSC resistance to anticancer drugs.

The anticancer drug doxorubicin (DOX) was used to evaluate drug resistance. As shown in Figure 10 (upper and lower panels), the cytotoxic effect of DOX on the cancer cells and CSCs increased in a concentration-dependent manner. Furthermore, drug resistance appeared more frequently in CSCs than in cancer cells. However, the cancer cell-killing effect decreased following PRKAR1A shRNA treatment (Figure 10A); cytotoxicity against CSCs increased in a concentration-dependent manner (Figure 10B).

Apoptosis assays were conducted using flow cytometry to validate the cytotoxic effects of DOX. A DOX concentration of 2 µM, determined by cell viability, was employed for the cancer cells, whereas the CSCs were treated with 4 µM DOX. As shown in Figure 11A, the shC+DOX treatment reduced the death of cancer cells compared to other groups. In contrast, increased CSC death was observed in the shC+DOX group (Figure 11B), which is consistent with the results shown in Figure 10.

In summary, these results suggest that the downregulation of PRKAR1A expression increases resistance to anticancer agents, leading to enhanced cancer cell-killing effects. Conversely, it reduces resistance to anticancer treatments in CSCs.

### 2.6. PRKAR1A shRNA Reduces Drug Resistance

PRKAR1A knockdown resulted in increased sensitivity to anticancer drugs, resulting in enhanced cell-killing effects against CSCs. Therefore, we evaluated the effect of PRKAR1A knockdown on the cytotoxic effects of anticancer drugs on drug-resistant cancer cells. The cells were repeatedly treated with 2 µM DOX three times every 48 h to generate drug-resistant cells. They were subsequently transfected with shPRKAR1A and treated with DOX at concentrations of 0, 1, 2, 4, and 5 µM for 24 h.

We observed the expression pattern of the multidrug resistance 1 (MDR1) gene to confirm whether the cells exposed to DOX had acquired resistance to anticancer drugs (Appendix A, and Figure 12A–D). After treating the cells which had developed anticancer drug resistance with shRNA, we evaluated how the cytotoxic effects on cancer cells changed following administration of the anticancer drug. As shown in Figure 12E,F, increased cell survival rates were observed in drug-resistant cells compared to non-resistant cells after anticancer drug treatment. For example, A549 cells treated with 4 µM DOX (Figure 10A) exhibited a 66.4% survival rate. However, the results of the same concentration treatment shown in Figure 12E revealed an 82.5% improvement in the survival rate following the development of drug resistance, indicating heightened chemoresistance in drug-resistant cells. Furthermore, the shC treatment significantly increased the survival rate of cancer cells. In contrast, the CSCs with acquired drug resistance exhibited increased survival rates when treated with DOX, whereas the shC treatment decreased their survival rates (Figure 12F). These results indicate that PRKAR1A enhances the cytotoxic effect of anticancer treatment on drug-resistant cells by modulating MDR1 expression.

### 2.7. Bioinformatic Survival Data Based on PRKAR1A Expression in Patients with Cancer

Our findings demonstrated that PRKAR1A exerts its effects on cells with different patterns in both cancer and cancer stem cells. Therefore, we used bioinformatic techniques to determine whether this gene could be used as a diagnostic marker in patients with cancer and compared the survival rates of patients with cancer from The Cancer Genome Atlas (TCGA). The clinical results confirmed PRKAR1A expression in various cancer types, including lung adenocarcinoma, lung squamous cell carcinoma, liver hepatocellular carcinoma, and breast cancer (Figure 13). Although PRKAR1A expression was not consistently high in the analyzed data of patients with cancer, it was highly expressed in patients with LIHC (Figure 13C).

When PRKAR1A was overexpressed in all cancer types, the survival period would begin to significantly decrease compared to than in patients with a low expression of this protein at a specific time point (Figure 13E–H). This indicates that PRKAR1A expression is associated with differences in patient survival rates. However, further research is required to elucidate the effect of reduced PRKAR1A expression on cancer aggressiveness. Nevertheless, the analysis of overall survival based on PRKAR1A expression revealed a consistent trend in the present study.

## 3. Discussion

The rise in cancer incidence, a perilous disease with high mortality rates worldwide, has sparked increased attention towards cancer therapy, leading to active research across various fields [1,2]. Despite these efforts, the inherent challenges associated with cancer, including the difficulty in achieving a complete cure, targeting cancer stem cells, and simultaneously diagnosing multiple types of cancer, remain unresolved [4,5,6,13]. Furthermore, cancer treatments often have significant side effects, such as acute or chronic toxicity of anticancer drugs, inability to exclusively target tumors (posing a risk of attacking normal cells), and induction of immune responses [7]. To overcome these limitations, this study aimed to conduct proteomic analyses to identify cell surface biomarkers that are commonly expressed in various cancer types. These biomarkers could serve as diagnostic tools and therapeutic agents for various cancer types.

In this study, we targeted three cancer types based on their predicted high incidence, consistently elevated mortality rates, and associated treatment challenges. First, we chose lung cancer, because of the increasing number of patients with post-COVID-19 complications and the high mortality rates across both sexes [14]. Second, liver cancer was included in our study, as its rising incidence is associated with the growing prevalence of obesity and irregular lifestyle choices [15]. Finally, we focused on breast cancer, which has high incidence rates among women, characterized by challenging treatment dynamics due to frequent metastasis and the presence of substantial lymph nodes in its vicinity [16]. By concentrating on the development of biomarkers specific to lung, liver, and breast cancers and exploring CSCs as potential candidates for anticancer treatment and diagnosis, our study aimed to provide valuable insights that could pave the way for more effective and targeted interventions in the battle against these challenging malignancies.

We initiated a proteomic analysis to identify new biomarker proteins in lung, liver, and breast cancer cells as well as CSCs (Figure 1). We selected the protein PRKAR1A, which is expressed on the cell surface, for potential applications as a diagnostic marker and target molecule during drug development. Therefore, we identified PRKAR1A as a potential therapeutic agent. This protein subunit, encoded by the PKA gene and part of the cAMP-dependent protein kinase A [17], emerged as a candidate biomarker. However, there has been little research on this protein. In addition, there are almost no functional studies on CSCs.

Despite the recognized role of cAMP in regulating various metabolic processes, including cell proliferation, differentiation, and apoptosis, the specific influence of PRKAR1A on cancer remains inadequately studied, with conflicting results within existing research on different cancer types [17,18,19,20,21]. To elucidate the biological function of PRKAR1A, we generated a series of shRNAs, with shC being the most effective in all the cells examined (Figure 2).

Our subsequent assessment of the effect of PRKAR1A on cancer cells and CSCs focused on the key features of cancer growth and metastasis [3,22,33], including short-term cell proliferation and long-term colony formation, both of which are regulated by PRKAR1A in opposite directions in these cell types (Figure 3 and Figure 4). Understanding the opposing trends induced by PRKAR1A involved examining the alterations in the cell cycle based on cyclin D1 expression (Figure 5). Cyclin D1, a crucial cell cycle regulator which is often overexpressed in cancer, promotes entry into the S phase and cancer growth [23,24]. The downregulation of PRKAR1A expression increased cyclin D1 expression and the S phase length of the cancer cells. Contrastingly, it decreased cyclin D1 expression and shortened the length of the S phase in CSCs. A similar phenomenon was observed in the cell mobility experiments of Figure 6, another key characteristic of cancer cells [3,33].

Taken together, the implications of these results are quite significant and the clinical application of PRKAR1A is quite limited because this protein functions oppositely in cancer cells and CSCs. Considering the clinical application of this protein, its initial use should be considered as a diagnostic biomarker. This gene is expressed in both cancer cells and CSCs, and, particularly, as seen in the results of Figure 13, its expression in clinical cancer patients is strongly associated with low survival rates. Therefore, considering its potential use as a prognostic biomarker for cancer progression, its use as a therapeutic agent may be limited to targeting CSCs.

To elucidate the fundamental biological functions mediating the opposing roles in cancer cells and CSCs, we examined the ERK pathway in cell signaling. This signal is a key pathway associated with cancer-related processes, such as cell proliferation, migration, differentiation, cell cycle regulation, and apoptosis [25]. Additionally, it plays a significant role in tumor formation and in the malignant progression of cancer [34,35]. Moreover, ERK phosphorylation is strongly associated with EMT [27]. EMT is a process in which cells lose cell–cell adhesion, gain mobility and invasiveness, and transform into mesenchymal-like stem cells, with the ability to reversibly transition between epithelial and mesenchymal states [28]. Given its implications in cancer malignancy and anticancer drug resistance, EMT is a crucial and well-recognized mechanism in cancer research [36]. Furthermore, EMT is associated with the acquisition of stemness, a feature attributed to stem cells, based on their ability to self-replicate and differentiate into various cell types [32,33]. Well-known stemness markers include *Sox2*, *Oct4*, and *Nanog*, which are highly expressed in CSCs [37]. These stemness markers are highly expressed in various cancer types, including lung, liver, and breast cancers [38,39]. Furthermore, they are closely associated with cancer initiation, recurrence, increased metastasis caused by enhanced EMT, resistance to multiple anticancer drugs, and poor prognosis [40,41,42]. Several mechanisms regulate stemness, including the Wnt/β-catenin, Notch, JAK/STAT, and PI3K/AKT/mTOR pathways [37]. However, studies investigating whether the ERK pathway regulates stemness are limited. Therefore, examination the ERK pathway and its association with EMT provides a crucial link between PRKAR1A, cancer malignancy, and stemness. This study elucidates how PRKAR1A affects EMT in cancer cells and MET in CSCs through ERK phosphorylation, establishing its role in cell behavior.

In this study, p-ERK expression increased in the cancer cells and decreased in the CSCs. Moreover, no change in p-ERK expression was observed after treatment with the ERK inhibitor PD98059, indicating that PRKAR1A directly regulates ERK phosphorylation (Figure 7). We further examined the reversible changes caused by ERK phosphorylation in the EMT mechanism. ERK phosphorylation induced EMT in the cancer cells, whereas it induced MET in the CSCs (Figure 8). The associated changes in the stemness markers (Figure 9) indicate that PRKAR1A regulates EMT and stemness through the ERK signaling pathway. Thus, the downregulation of PRKAR1A expression in cancer cells signifies the ERK phosphorylation-facilitated induction of EMT and increased stemness, elucidating the reasons behind the observed enhancements in cell proliferation and migration. Conversely, a decrease in ERK phosphorylation indicates MET and reduced stemness in CSCs, providing an explanation for the observed reductions in cell proliferation and migration. Thus, confirmation that the ERK pathway is associated with EMT and stemness underscores its relevance to signaling pathways.

Successful cancer treatment involves early detection and removal through conventional therapies. However, drug resistance increases over time, and further anticancer treatments become less effective. Specifically, it is well-known that CSCs exhibit resistance to anticancer drugs compared to cancer cells [6] and the epithelial-to-mesenchymal transition (EMT) program in cancer cells provides fundamental insights into anticancer drug resistance via an enhanced drug efflux ability. Also, CSCs exhibit diversity and plasticity, enabling them to differentiate into various cell types to survive targeted anticancer drug treatments. They possess mechanisms to evade attacks from the immune system and anticancer drugs, enhancing their resistance to treatment. Another reason why cancer stem cells are resistant to anticancer drugs is because of the tumor microenvironment (TME) and CSC niche. The TME comprises diverse components such as cytokine networks, chemokines, and growth factors, which can regulate CSCs’ self-renewal, angiogenesis, and remodeling of immunity. Consequently, the CSC niche, through pathways like Wnt/ß-catenin and Notch, controls the transcription of stemness markers like Nanog, Oct4, Sox2, and Sox9, maintaining stemness and allowing CSCs to evade anticancer drugs [43].

Therefore, considering the implications of EMT and stemness on chemoresistance [42], we evaluated how PRKAR1A contributes to drug resistance because its downregulation in CSCs resulted in MET in our findings. Therefore, we conducted assessments related to drug resistance and demonstrated that the knockdown of PRKAR1A led to an increase in the cytotoxicity of anticancer agents in normal CSCs and drug-resistant CSCs (Figure 10 and Figure 12).

In this regard, we generated drug-resistant cancer cell lines and CSCs to investigate whether decreased PRKAR1A levels can reduce chemoresistance in cells with inherent drug resistance. The confirmation of the downregulation of the drug-resistance gene MDR1 demonstrated that PRKAR1A induced chemoresistance in both the cancer cells and the CSCs through the regulation of MDR1 expression. MDR1, also known as ATP-binding cassette transporter C1, transports various therapeutic agents, including DOX, methotrexate, edatrexate, and daunorubicin, out of the cell membrane, thereby regulating chemoresistance [44,45]. When we compared the expression of MDR1 between drug-resistant and normal cell lines, we observed higher MDR1 expression levels in the drug-resistant cell lines. Furthermore, the baseline expression level of MDR1 was higher in the CSCs than in the cancer cells. These results are consistent with reports indicating that CSCs possess a higher resistance to anticancer agents than cancer cells. This was observed in a similar way to the preceding cell proliferation findings, suggesting that cancer cells and CSCs may possess different mechanisms of drug resistance. This insight implies that PRKAR1A can be applied more widely to address drug resistance and improve therapeutic effects.

The evaluation of PRKAR1A expression in patients with cancer using TCGA data (Figure 13) revealed low expression of this protein in certain cancer types. Higher expression levels were correlated with lower survival rates, emphasizing the need for tailored analyses of biomarker development. Our results confirm a strong association between low survival rates and the expression of this protein. Therefore, we consider that the development of therapies utilizing this protein should be approached separately for cancer cells and CSCs.

There is growing interest in the discovery of biomarkers that can offer new directions for anticancer drugs to address the continually rising incidence and mortality rates of cancer. Thus, this study focused on the discovery and validation of novel biomarkers with broad applicability in various cancer types. PRKAR1A, identified through a proteomic analysis, emerged as a promising cell surface-expressed biomarker capable of regulating cancer’s biological functions, EMT, and stemness through the ERK signaling pathway, consequently reducing the cytotoxic effects associated with resistance to anticancer agents. Targeting PRKAR1A could offer a strategic approach for overcoming the limitations of conventional cancer treatment and pave the way for new therapeutic approaches.

## 4. Materials and Methods

### 4.1. Cell Culture

#### 4.1.1. Cancer Cells

The cell lines A549 (human lung cancer cell), Huh7 (human liver cancer cell), MCF7 (human breast cancer cell), and UCB-MSC (umbilical cord blood-derived mesenchymal stem cell) were all purchased from the American Type Culture Collection (ATCC, Manassas, VA, USA). The A549 and Huh7 cell lines were cultured in Dulbecco’s Modified Eagle’s Medium (DMEM) (Hyclone, Logan, UT, USA) containing 10% fetal bovine serum (FBS) (Hyclone, UT, USA) and 1% penicillin-streptomycin (PS) (Hyclone, UT, USA). The MCF7 cells were cultured in an RPMI 1640 medium (Welgene, Gyeongsangbuk-do, Republic of Korea) containing 10% FBS and 1% PS. The UCB-MSC cells were cultured in a stem cell-conditioned basal medium of KSB-3 (Kangstem Biotech, Seoul, Republic of Korea) and were added to KSB-3 supplements with 10% FBS. All the cell cultures were incubated at 37 °C in a humidified incubator with 5% CO_2_.

#### 4.1.2. Cancer Stem Cells (CSCs)

The cell lines lung cancer-derived stem cell (LCSC), liver cancer-derived stem cell (LiCSC), and breast cancer-derived stem cell (BCSC) were cultured in DMEM/F-12 Nutrient Mixture Ham (DMEM/F-12) (Welgene, Gyeongsan-si, Gueongsangbuk-do, Republic of Korea) containing 10% FBS, 1% PS, 5 µg/mL insulin (Invitrogen, Carlsbad, CA, USA), 20 ng/mL EGF (Gibco, Gaithersburg, MD, USA), 20 ng/mL b-FGF (Gibco), and 1% B27 (Invitrogen, CA, USA). The cancer cell lines were cultured in Costar^®^ 6-well Clear Flat Bottom Ultra-Low Attachment Multiple Well Plates (Corning, NY, USA), with an appropriate medium. The cancer cells were seeded at a density of 2 × 10^4^ cells/well to make CSCs. All the CSCs were sub-cultured on the seventh day of culture and incubated for two weeks (Figure 1.). All the cell cultures were incubated at 37 °C in a humidified incubator with 5% CO_2_ [46].

### 4.2. Mass Spectrometry Analysis

The proteomic analysis from cancer cells and CSCs was conducted similarly to our previous paper [47]. In brief, the process involved the following steps: Cancer cells and CSCs were lysed in a 30 µL buffer at 4 °C, and the resulting supernatant underwent tryptic digestion after centrifugation. The protein concentration was determined using Coomassie Plus Assay Reagent. The lysate proteins were treated with DTT and IAA, followed by trypsin digestion. A cleanup with an MCX cartridge involved equilibration, washing, and elution steps using specific solutions. The eluate was dried, and peptides were either extracted with formic acid for LC injection or stored at −20 °C before analysis. The samples underwent separation through online reversed-phase chromatography using Thermo Scientific equipment (Thermo Fisher Scientific, Waltham, MA, USA). This included an Easy nano LC II autosampler, a peptide trap EASY-Column, and an analytical EASY-Column. Electrospray ionization was performed with a nano-bore stainless steel online emitter. The chromatography system was coupled with an LTQ Velos Orbitrap mass spectrometer featuring an ETD source. A data-dependent switching mode was applied to enhance peptide fragmentation, and protein identification utilized Sorcerer 2 against the 2014 UniProt human DB. The set tolerances included 1.0 Da for fragment mass, 25 ppm for peptide mass, and a maximum of two missed cleavages. Result filters were applied, considering a minimum number of peptides per protein, with static and variable modifications. The processed datasets were transformed to .sf3 files using the Scaffold 3 program.

### 4.3. RNA Extraction and Conventional Polymerase Chain Reaction (PCR)

At 48 h post transfection, the cells were harvested, and the total RNA was extracted according to the manufacturer’s manual using TRIzol (Invitrogen, CA, USA). Then, it was quantified by Nanodrop^TM^ (Thermo Fisher Scientific, Waltham, MA, USA). The extracted RNA was converted into complementary DNA (cDNA) with 2X RT Pre-Mix (Solgent, Daejeon, Republic of Korea). The synthesized cDNA was subjected to conventional PCR using a 2X Taq PCR Pre-Mix (Solgent, Daejeon, Republic of Korea) according to the manufacturer’s protocols. All the samples synthesized by conventional PCR were confirmed by separation through 2% agarose gel electrophoresis (Vivantis, Molecular Biology Grade, CA, USA) in a TAE buffer. Gel images were taken using a chemiluminescence detection system (VilberLourmat, Everhardzell, Germany). The primer sequences for performing PCR are shown in Table 1.

### 4.4. Transfection

PRKAR1A short hairpin RNA (shRNA) and scrambled shRNA (shScr) were constructed in the psi-LVRU6GP vector (Genecopoeia, Rockville, MD, USA) using the same sequences (NM_002734.4). The PRKAR1A shRNA targeting construct was 5′-GCATAACATTCAAGCGCTGCT-3′. Cancer cells were seeded 1.5 × 10^5^ cells/well, and CSCs were seeded 2 × 10^5^ cells/well on a six-well plate and then transfected with the psi-LVRU6GP vector containing shRNA-PRKAR1A or shRNA (Genecopiea, MD, USA) using the Lipofectamine^TM^ 3000 Transfection Reagent (Invitrogen, CA, USA) in an FBS-reduced medium according to the manufacturer’s instructions. Scramble shRNA (shScr) in the same plasmid was used as the positive control and cell-only groups were used as the negative control. At 4 h post transfection, the medium changed to a completely fresh medium. After the transfection, the RNA was extracted, and we conducted a conventional PCR to validate the expression of PRKAR1A.

### 4.5. Cell Proliferation Assay

Cancer cells were seeded onto six-well plates at 1.5 × 10^5^ cells/well, and the CSCs were seeded onto six-well plates at 2 × 10^5^ cells/well. Then, shRNA series were transfected into each cell line as described, and then, 4 h later, the cells were seeded in triplicate onto a 96-well plate at a density of 8 × 10^3^ cells/well. After 48 h, cell viability was measured by 3-(4,5-Dimethylthiazol-2-yl)-2,5-diphenyltetrazolium bromide (MTT assay) (Sigma-Aldrich, Poole, UK) depending on the time needed to confirm cell proliferation. After removing the culture medium, 100 µL of MTT reagent (2 mg/mL) was added to each well and incubated at 37 °C in a CO_2_ incubator for 4 h. Following the incubation period, the MTT reagent was removed, and 100 µL of dimethyl sulfoxide (DMSO; Sigma-Aldrich, St. Louis, MO, USA) was added to each well to react with the generated formazan for 20 min. Absorbance was measured at 540 nm using a microplate reader (Versamax microplate reader; Molecular Devices Corp., Sunnyvale, CA, USA). Cell proliferation was compared relative to the 0 h time point.

### 4.6. Cell Colony Formation Assay and Crystal Violet Stain

Cancer cells and CSCs were seeded at 1 × 10^3^ cells/well onto a six-well plate with an appropriate medium and then incubated for 10 days. The cell culture medium was replaced every 2 days. When a visible colony appeared, the cells were washed three times in PBS. Next, the cells were fixed with methanol for 20 min and stained with 0.1% crystal violet solution for 15 min in the dark, at room temperature. After staining and visually confirming the formation of colonies, the crystal violet solution in the colonies was dissolved by 100% methanol. The dissolved crystal violet was then distributed into a 96-well plate with 100 µL in each well and measured for absorbance at 570 nm using a microplate reader (Versamax microplate reader, Molecular Devices Corp., CA, USA). The measured values were compared to the untreated cells, which were considered to have a value of 100%, to assess the relative differences in the other samples.

### 4.7. Cell Cycle Assay by Flow Cytometry

Cancer cells and CSCs after being transfected with shPRKAR1A for 48 h were collected and washed with cold PBS three times. And then, the cells were fixed with 70% ethanol at 4 °C for 2 h. After being washed with cold PBS three times, the cells were stained with a propidium iodide (PI) solution (Thermo Fisher Scientific, Waltham, MA, UAS) at 4 °C overnight. The cell cycles were analyzed with a Novocyte Flow Cytometer (ACEA Bioscience Inc., San Diego, CA, USA).

### 4.8. Wound Healing Scratch Assay

After transfection as described, the cancer cells were seeded at 6 × 10^4^ cells/well and at 8 × 10^4^ cells/well for the CSCs onto 24-well plates. When the confluence of cells prepared in each group reached 80%, they were scraped to make them group uniformly wide with a 200 µL sterile pipette tip. Photographs were taken 0 and 12 h after the scratch’s creation with a microscope (Nikon eclipse Ts2R). The ratio of cell movement in the 12 h sample compared to the 0 h sample was quantitatively analyzed by the image J program.

### 4.9. Western Blot Analysis

Protein extraction was performed for groups subjected to transfection with shC for 48 h, followed by a 1 h treatment with PD98059 (R&D systems, Minneapolis, MN, USA) at a concentration of 15 µM, and compared with the group that did not receive the treatment. The samples were harvested and washed with cold PBS two times. The cell pellets were lysed using the EzRIPA Lysis Kit (RIPA buffer and inhibitors) (DAWINBIO Inc., Gyeonggido, Republic of Korea) for 15 min on ice. The lysates were centrifuged at 13,000 RPM for 15 min at 4 °C, and the protein supernatant was measured using the BCA protein assay reagent (Thermo Fisher Scientific, Waltham, MA, USA). Equal amounts of proteins (15 µg) were separated by 10% sodium dodecyl sulfate polyacrylamide gel electrophoresis (SDS-PAGE). The proteins were transferred onto an Immobilon^®^-P PVDF membrane (Millipore Co., MA, USA) and blocked with 3% skim milk for 1 h. Then, the membranes were incubated with 3% skim milk containing primary antibodies at 4 °C overnight. The primary antibodies against β-actin (1:1000, sc-4778, Santa Cruz, CA, USA), E-cadherin (1:1000, 3195, Cell Signaling Technology (CST), Danvers, MA, USA), N-cadherin (1:500, 14215S, CST), and p-ERK (1:1000, sc-7383, Santa Cruz.) were used. After incubation, the membranes were incubated for 1 h at room temperature with a horse radish peroxidase (HRP)-conjugated secondary antibody. For the detection of the secondary antibodies targeting β-actin, N-cadherin, and p-ERK, the HRP-conjugated anti-mouse IgG antibody (1:10,000, 31,430, Invitrogen, Carlsbad, CA, USA) was used, whereas the HRP-conjugated anti-rabbit IgG antibody (1:10,000, 7074, CST) was used for capturing E-cadherin. The protein band were developed with a chemiluminescent ECL reagent (Thermo Fisher, Waltham, MA, USA) using an enhanced chemiluminescence detection system (VilberLourmat, Everhardzell, Germany). All the primary antibodies were normalized to β-actin. The quantification of protein bands was performed using the Image J software.

### 4.10. Drug Resistance Assay

After transfection as described, cancer cells and CSCs were seeded in a 96-well plate at a density of 8 × 10^3^ cells/well. Subsequently, the anticancer drug doxorubicin (Doxorubicin hydrochloride; Sigma Chemical, St. Louis, MO, USA) was administered at concentrations of 1, 2, 4, and 5 µM for 24 h. Following treatment, the medium was replaced with a fresh medium, allowing a 48 h recovery period for the cells. A subsequent MTT assay was conducted to measure cell viability. After media removal, 100 µL of MTT (Sigma) reagent was added and incubated for 4 h at 37 °C. The formazan crystals formed by the cells were dissolved in DMSO (Sigma-Aldrich) and measured at 540 nm using VerxaMax (Microplate Reader, Molecular Devices, Sunnyvale, CA, USA). The cells treated with different concentrations of doxorubicin were quantitatively compared to the values of the untreated cells, and the results were represented graphically.

### 4.11. Generation of Chemoresistant Cell Lines

After treating the cells with the anticancer drug doxorubicin at a concentration of 2 µM for 24 h, a recovery period of 48 h was provided by changing the medium to a fresh medium. This process was repeated three times, and then the morphological changes in the surviving cells were examined using Zoe (Bio-Rad Laboratories, Hercules, CA, USA). Following the development of anticancer drug resistance in the cells, shPRKAR1A was transfected as described previously. Subsequently, conventional PCR and MTT assays were conducted.

### 4.12. Apoptosis Assay

Apoptosis was detected using the FITC-Annexin V Apoptosis Detection Kit 1 (BD Bioscience, Bedford, MA, USA) according to the manufacturer’s protocol. Samples were harvested and washed twice with cold PBS and re-suspended in 1× binding buffer at a concentration of 1 × 10^6^ cells/mL. The cells were transferred to 100 µL of the solution (1 × 10^5^ cells/mL), and 5 µL of FITC-Annexin V and 5 µL propidium iodide (PI) were added to stain the cells for 15 min in the dark, at room temperature. Next, 400 µL of 1× binding buffer was added to each sample. Finally, the apoptotic levels were analyzed by flow cytometry (Novo Cyte flow cytometer, ACEA Bioscience Inc., Santa Clara, CA, USA). The data were analyzed using the Novoexpress software version 1.2.5 (ACEA Biosciences Inc., USA).

### 4.13. Data Acquisition and Preprocessing

Expression and survival data for lung adenocarcinoma (LUAD), lung squamous cell carcinoma (LUSC), liver hepatocellular carcinoma (LIHC), and breast invasive carcinoma (BRCA) were obtained from The Cancer Genome Atlas (TCGA) via the OncoDB database. The data were subjected to preprocessing, which included the normalization of expression values and the confirmation of data integrity. The final dataset consisted of normalized expression values for the PRKAR1A gene and the corresponding patient survival information, including time-to-event and vital status.

### 4.14. Statistical Analysis of Expression Data

A differential expression analysis was performed to compare the PRKAR1A expression levels between cancerous and normal tissue samples. For this purpose, a two-sample *t*-test was employed, assuming unequal variances. The *t*-test was conducted to ascertain the significance of expression differences for PRKAR1A between the cancer samples and the matched normal controls.

### 4.15. Survival Analysis

A survival analysis was conducted to investigate the association between PRKAR1A expression levels and patient prognosis. Kaplan–Meier survival curves were plotted to visualize differences in survival probabilities between high and low PRKAR1A expression groups. The survival curves were compared using the log-rank test to evaluate the statistical significance of differences in the survival times between groups. The Cox proportional hazards model was utilized to estimate the hazard ratios, providing a measure of the effect size of PRKAR1A expression on survival. The model was adjusted for potential confounders, including age, sex, and cancer stage. Hazard ratios with 95% confidence intervals were calculated to assess the relative risk of mortality associated with PRKAR1A expression levels. A differential expression analysis was performed to compare the PRKAR1A expression levels between cancerous and normal tissue samples. For this purpose, a two-sample *t*-test was employed, assuming unequal variances. The *t*-test was conducted to ascertain the significance of expression differences for PRKAR1A between the cancer samples and the matched normal controls.

### 4.16. Software

All statistical analyses were performed using R statistical software (version 3.6.3). The survival package was used for the survival analyses, and the ggplot2 package was employed for generating boxplots and Kaplan–Meier plots. *p*-values of less than 0.05 were considered indicative of statistical significance.

### 4.17. Statistical Analysis

All the experiments were performed at least three times, and the measured data were calculated as the mean ± standard error of the mean (SEM) and presented as a graph. The significance test between groups was analyzed by one-way ANOVA. Statistical differences were indicated in the figures. The significance levels were as follows: non-significant (ns) > 0.05, * *p* < 0.05, ** *p* < 0.02, and *** *p* < 0.01. For the statistical analysis, the SPSS statistics software for Windows, Version 18 (SPSS Inc., Chicago, IL, USA), was used.

## 5. Conclusions

This study is the first report evaluating the potential of newly discovered anticancer-targeting biomarkers as theragnostic molecules. Identified through a proteomic analysis, these biomarkers have the unique capability to target both cancer cells and CSCs derived from various cancer types. This study focused on the identification of common biomarkers expressed on the surfaces of diverse cancer cells with the objective of simultaneously targeting cancer cells and CSCs. The identified biomarker, PRKAR1A, exhibited expression in both cancer cells and CSCs, showcasing its ability to regulate EMT and stemness through the ERK signaling pathway. Additionally, PRKAR1A demonstrated the capacity to influence the sensitivity of cancer cells and CSCs to anticancer agents, thereby impacting their cell-killing effects. The clinical outcomes associated with elevated levels of PKRAR1A underscore its significance as an unfavorable prognostic factor in cancer, emphasizing the critical need for further research on PRKAR1A in the context of cancer. In conclusion, this study proposes PRKAR1A as a promising diagnostic and therapeutic marker for cancer, highlighting its functional role as a biomarker which could contribute to the development of novel strategies in anticancer treatments.

## Data Availability

Data are contained within the article and Appendix A.

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
