# Peer review of "Dualistic Effects of PRKAR1A as a Potential Anticancer Target in Cancer Cells and Cancer-Derived Stem Cells"

_ijms, 2024, doi:10.3390/ijms25052876_

Round 1

Reviewer 1 Report

Comments and Suggestions for Authors

Kyung-gon and Pyung-Hwan et al. submitted the manuscript entitled: Dualistic effects of PRKAR1A as A Potential Anticancer Target in Cancer Cells and Cancer-Derived Stem Cells, in which they investigated on different role of PRKAR1A in cancer cell and cancer stem cells. They found downregulation of PRKAR1A led to increase of cancer cell proliferation (but decrease in cancer stem cells). The authors also tried to investigate the mechanism and found PRKAR1A has relation to CyclinD1/CDK and ERK pathway. I have some concerns as follows.

1. Figure 1: The authors are suggested to add statistical analysis between each cancer cell and its corresponding cancer stem cell (eg. A549 and LCSC). This expression difference might be helpful in further mechanism investigation.

2. Figure 2: Please define shScr (should be negative control?).

Besides, for all the related figures, please also label out statistic difference between each blank control and shScr.

3. Figure 3: Label out the statistic difference for shScr group and blank control, especially for MCF7 group.

4. The authors stated “the altered cell proliferation rate in each cell is considered to be influenced by changes in proteins associated with the cell cycle.” Please include reference. Similar suggestion for statement in line 204: “To address this, we focused on the representative signaling 204 pathway, ERK, which is associated with many intracellular functions.” From current version of manuscript, readers might be confused why the authors come up with verification of these pathways.

5. Figure 10 and Figure 12, trend of Dox sensitivity on normal cancer cell, normal stem cancer cell, drug-resisted cancer cell and drug-resisted cancer stem cell induced by shC are not prominent. From current version of data, I cannot accurately tell the effect of shC in enhancing drug sensitivity of these cell lines. The author can considering applying a fixed dose (highest dose that cells can tolerate) of Dox in these cells with different time span to see the time-dependency.  

Additionally, it seems that original (not Dox resistant) A549, Huh7 and MCF7 used in this work also show resistant ability against Dox.

6. Section 2.7: considering the facts that 1) cancer stem cell only accounts for 1% in tumor and 2) in this work, PRKAR1A knockdown actually enhanced cancer cell proliferation, more discussion should be involved in whether PRKAR1A knockdown can truly lead to tumor regression and bring clinical benefit.

7. Some small typos exist: line 36, “akin” should be “a kin”; line 86, “Nexy” should be “Next”; line 284: “tesistance” should be “resistance”; line 286: “cnacer” should be “cancer”. This is not a completed form of all the typos and the authors should double-check the whole manuscript.

Comments on the Quality of English Language

Language can be understood.

Some typos were mentioned in the Comments and Suggestions for Authors.

Author Response

Point-by-point response to the Reviewer #1

  1. [Comment] Figure 1: The authors are suggested to add statistical analysis between each cancer cell and its corresponding cancer stem cell (eg. A549 and LCSC). This expression difference might be helpful in further mechanism investigation.

[Answer] We appreciate your comment very much. When we conducted statistical analysis between cancer cells and cancer stem cells (CSCs), no significant differences were observed in A549 and LCSC, Huh7 and LiCSC, MCF7 and BCSC (ns P > 0.05). This suggests that PRKAR1A is expressed at similar levels in both cancer cells and cancer stem cells, showing a significant difference compared to normal stem cells such as UCB-MSC. Thus, PRKAR1A is expected to function as a substance that can specifically diagnose cancer. Furthermore, as advised by the reviewer, it is anticipated that such expression differences will be adequately utilized in future mechanistic studies of cancer cells to elucidate the significant biological functional variances of this protein in cancer and cancer stem cells.

  1. [Comment] Figure 2: Please define shScr (should be negative control?). Besides, for all the related figures, please also label out statistic difference between each blank control and shScr.

[Answer] Thank you for your kind comment. In all study, shScr was used as positive control compared  to shC, and cell only (eg. A549, Huh7, or MCF7) was used as the negative control. We have added the definition of shScr under Transfection in Materials & Methods section (line 595 on page 24). Furthermore, as your comment, we illustrated the statistical differences between each cell only (blank control) and shScr in all figures. ns P > 0.05, *P < 0.05, **P < 0.02, and ***P < 0.01 compared with cell only versus shScr groups.

  1. [Comment] Figuree 3: Label out the statistic difference for shScr group and blank control, especially for MCF7 group.

[Answer] Thank you for your kind comment. We represented the statistical differences between all cell only (blank control) and shScr groups for shScr in all results, as per your comment #2. ns P > 0.05, *P < 0.05, **P < 0.02, and ***P < 0.01 compared with cell only versus shScr groups.

  1. [Comment] The authors stated “the altered cell proliferation rate in each cell is considered to be influenced by changes in proteins associated with the cell cycle.” Please include reference. Similar suggestion for statement in line 204: “To address this, we focused on the representative signaling pathway, ERK, which is associated with many intracellular functions.” From current version of manuscript, readers might be confused why the authors come up with verification of these pathways.

[Answer] Thank you for raising this point. As your comment, we added the reference [23] for the statement ‘the altered cell proliferation rate in each cell is considered to be influenced by changes in proteins associated with the cell cycle.’ in line 158 on page 6. And to prevent confusion among readers, the sentence line 204 in question has been revised as follows :

“Based on our previous data confirming that PRKAR1A regulates cell proliferation and migration, we sought to determine whether these effects associated with ERK phosphorylation in the ERK signaling pathway, well-known for the representative intracellular signaling pathway [25].” Line 206 on page 8. The sentence that has been revised according to your advice is highlighted in red

  1. [Comment] Figure 10 and Figure 12, trend of Dox sensitivity on normal cancer cell, normal stem cancer cell, drug-resisted cancer cell and drug-resisted cancer stem cell induced by shC are not prominent. From current version of data, I cannot accurately tell the effect of shC in enhancing drug sensitivity of these cell lines. The author can considering applying a fixed dose (highest dose that cells can tolerate) of Dox in these cells with different time span to see the time-dependency. 

[Answer] Thank you for pointing this out. In the results of Fig. 10 and 12, all tested cells exhibited a concentration-dependent cytotoxicity to the anticancer drug. To help in understanding reviewers' question, we have organized the cell survival values for each cell type in a tabular format. In the table, the survival values of each cell group tend to be lower when in the condition of normal cancer cells, but increase upon becoming drug-resistant cancer cells.

Cancer cells

Drug-resistant cancer cells

DOX

0 µM

1 µM

2 µM

4 µM

5 µM

0 µM

1 µM

2 µM

4 µM

5 µM

A549

100

89.1

81.7

66.4

47.1

100

98.2

94.4

82.5

80.6

shScr

88.2

82.3

63.2

45.9

98.7

93

81.1

79.3

ShC

94

87.3

73.2

64.1

99

92.2

89.4

86.6

Cancer cells

Drug-resistant cancer cells

DOX

0 µM

1 µM

2 µM

4 µM

5 µM

0 µM

1 µM

2 µM

4 µM

5 µM

Huh7

100

76.5

69

57.7

49.8

100

94.4

88.9

71.4

62.5

shScr

79.3

69.2

56

51.7

95.5

87.3

71

60.9

ShC

90.8

83.9

70.8

60.7

94

88.4

80.7

71.1

Cancer cells

Drug-resistant cancer cells

DOX

0 µM

1 µM

2 µM

4 µM

5 µM

0 µM

1 µM

2 µM

4 µM

5 µM

MCF7

100

92.5

78.3

68.3

58.5

100

96.5

92.6

81.8

73.5

shScr

90.5

77.8

65.8

61.6

94.7

91

81.7

72.5

ShC

95.9

86.7

79.4

72.9

99.8

95.6

89.8

80.5

Cancer cells

Drug-resistant cancer stem cells

DOX

0 µM

1 µM

2 µM

4 µM

5 µM

0 µM

1 µM

2 µM

4 µM

5 µM

LCSC

100

96.6

86.2

76.5

68.8

100

92.4

88.1

83.9

82.3

shScr

96.9

85.3

74.7

68.4

91

90.7

84.3

81.6

ShC

89.6

78.9

64.6

58.1

91.7

81.6

70.8

66.2

Cancer cells

Drug-resistant cancer stem cells

DOX

0 µM

1 µM

2 µM

4 µM

5 µM

0 µM

1 µM

2 µM

4 µM

5 µM

LiCSC

100

94.6

90.6

78

69.1

100

95.7

91.4

85.9

76.7

shScr

95.2

89.6

77.1

67.4

94.2

92

82.5

73.4

ShC

86.5

77.5

66.9

59.6

93.3

83.9

71

69.9

Cancer cells

Drug-resistant cancer stem cells

DOX

0 µM

1 µM

2 µM

4 µM

5 µM

0 µM

1 µM

2 µM

4 µM

5 µM

BCSC

100

95.9

90.5

82.9

68.3

100

90

88.9

85.3

81.6

shScr

94.9

87.3

83

66.5

90.7

87.4

83.6

80.5

ShC

90.3

82.6

70.4

60.7

88.6

87

70.9

71.6

While this trend difference is observed in CSCs, the difference in drug resistance is not as pronounced as in normal cancer cells. However, when shC was treated to CSCs, it still exhibited the best anticancer cell killing ability compared to other groups, whether in normal CSCs or drug-resisted CSCs. Therefore, induction of PRKAR1A down-regulation could serve as a promising target for CSC therapy.

  1. [Comment] Section 2.7: considering the facts that 1) cancer stem cell only accounts for 1% in tumor and 2) in this work, PRKAR1A knockdown actually enhanced cancer cell proliferation, more discussion should be involved in whether PRKAR1A knockdown can truly lead to tumor regression and bring clinical benefit.

[Answer] We appreciate your very important comment. As mentioned by the reviewer, PRKR1A functions oppositely in cancer cells and CSCs. Considering the clinical application of this gene, its initial use should be considered as a diagnostic biomarker. This gene is expressed in both cancer and CSCs, and particularly, as seen in the results of Figure 13, its expression in clinical cancer patients is strongly associated with low survival rates. Therefore, considering its potential use as a prognostic biomarker for cancer progression, its use as a therapeutic agent may be limited to targeting cancer stem cells. This information has been added to Discussion section. The modified parts of Discussion section are indicated in red.

  1. [Comment] Some small typos exist: line 36, “akin” should be “a kin”; line 86, “Nexy” should be “Next”; line 284: “tesistance” should be “resistance”; line 286: “cnacer” should be “cancer”. This is not a completed form of all the typos and the authors should double-check the whole manuscript.

[Answer] Thank you for your kind comment. We have corrected all the typos you mentioned and have reviewed the entire manuscript again to identify any further errors.

Reviewer 2 Report

Comments and Suggestions for Authors

In this manuscript, Baek et al. presented a study on finding new biomarkers for anticancer therapy. The experiments are well-designed and the results are answering their questions. I have several concerns for this paper before consideration of publication.

1. The discussion looks descriptive to me, and presents like a summary of the results. What's the explanation of the results? Have the authors compared the findings with other studies'? What's the unexpected findings and limitations of the methods? Any weakness in study design? 

2. Language needs more work for clarity.

3. Readers would want to get more information from the paper on how cancer stem cells get chemoresistant mechanistically, and how that's related to the results here.  

4. Figure 13 was not cited correctly. 

5. Actual p-values need to be added in figure legends, instead of appearing as a range.

Comments on the Quality of English Language

English is easy to understand. However, language mechanics needs more work and there is redundancy in texts and between sections. 

Author Response

  1. [Comment] The discussion looks descriptive to me, and presents like a summary of the results. What's the explanation of the results? Have the authors compared the findings with other studies'? What's the unexpected findings and limitations of the methods? Any weakness in study design? 

[Answer] We appreciate your comment very much. As the reviewer's advice, we have revised the Discussion section to focus more deeply on the significance of the results. The overall summary of our research findings is as follows:

We selected the three most prevalent and highly fatal types of cancer, aiming to identify common cell surface biomarkers (PRKAR1A) expressed in these cancers for the treatment of cancer stem cells (CSCs). Through our biological studies on cancer cells and CSCs, we have demonstrated their divergent biological functions. While this protein has been previously reported in lung cancer cells (Scientific Reports 2016 6:39630, DOI: 10.1038/srep39630), its biological function in CSCs has not been reported. Therefore, although we initially planned to develop a therapy targeting both cancer and CSCs using this common biomarker, unexpected contrasting results were observed: the inhibition of this protein's expression induces cell proliferation in cancer cells while decreasing it in CSCs. Consequently, this substance could serve as an indicator for predicting the prognosis of cancer. Our results in Fig. 13 confirm a strong association between low survival rates and the expression of this protein, reinforcing this fact. Therefore, we believe that the development of therapies utilizing this protein should be approached separately for cancer cells and CSCs. Furthermore, we plan to investigate the expression of this protein in a broader range of cancer and CSC types.

  1. [Comment] Language needs more work for clarity.

[Answer] Thank you for your kind comment. We have thoroughly reviewed the entire manuscript once again and corrected all the typos. Also, we received a review of the English level from an English editing company. As such, we have attached the proof of correction documentation.

  1. [Comment] Readers would want to get more information from the paper on how cancer stem cells get chemoresistant mechanistically, and how that's related to the results here.

[Answer] Thank you for pointing this out. CSCs are well known for its inherent resistance to anticancer agents (Stem cells Int 2018 2018:5416923, doi: 10.1155/2018/5416923). Cancer stem cells possess various factors contributing to their resistance to anticancer drugs and the epithelial-to-mesenchymal transition (EMT) program in cancer cells provides fundamental insights into anticancer drug resistance via enhanced drug efflux ability. Also, CSCs exhibit diversity and plasticity, enabling them to differentiate into various cell types to survive targeted anticancer drug treatments. They possess mechanisms to evade attacks from the immune system and anticancer drugs, enhancing their resistance to treatment. Another reason cancer stem cells are resistant to anticancer drugs is because of the tumor microenvironment (TME) and CSC niche. The TME comprises diverse components such as cytokine networks, chemokines, and growth factors, which can regulate CSCs' self-renewal, angiogenesis, and remodeling of immunity. Consequently, the CSC niche, through pathways like Wnt/ß-catenin and Notch, controls the transcription of stemness markers like Nanog, Oct4, Sox2, and Sox9, maintaining stemness and allowing CSCs to evade anticancer drugs [Cell Commun Signal 2021 19(1):19, doi: 10.1186/s12964-020-00627-5; Life Sci 2019 Oct 1:234:116781. doi: 10.1016/j.lfs.2019.116781]. In our study, we evaluated how PRKAR1A contributes to drug resistance because its downregulation in CSCs resulted in mesenchymal-to-epithelial transition (MET) in our findings on the epithelial-to-mesenchymal transition (EMT) in CSCs. Therefore, we conducted assessments related to drug resistance, and demonstrated that knockdown of PRKAR1A led to an increase in the cytotoxicity of anticancer agents in normal CSC and drug-resisted CSC with the increased MDR1 expression (Fig. 10 and 12). This information added to Discussion section and are indicated in red.

  1. [Comment] Figure 13 was not cited correctly. 

[Answer] We appreciate your comment. As the reviewer's comment, we reconfirmed citations pertaining to Fig. 13 and revised in manuscript (line 363 and 366 on page 19).

  1. [Comment] Actual p-values need to be added in figure legends, instead of appearing as a range.

[Answer] Thank you for your kind comment. As suggested, we added the p-values comparing the representative shScr and shC in all figure legends.

Round 2

Reviewer 1 Report

Comments and Suggestions for Authors

The authors have well addressed on all the issues.